# Usability of Mixed Reality for Naloxone Training: Iterative Development and Field Testing of ReviveXR

**DOI:** 10.3390/healthcare13121449

**Published:** 2025-06-17

**Authors:** Wasantha Jayawardene, Roy Magnuson, Chesmi Kumbalatara, Matthew Kase, Amy Park, Alana Goodson, Scott Barrows, Rebecca Bolinski, Joanna Willett

**Affiliations:** 1School of Human Sciences, Southern Illinois University, Carbondale, IL 62901, USA; chesmi.kumbalatara@siu.edu (C.K.); amy.park@siu.edu (A.P.); 2Wonsook Kim College of Fine Arts, Illinois State University, Normal, IL 61790, USA; rdmagnu@ilstu.edu; 3Design Lab, OSF HealthCare Innovation, Peoria, IL 61603, USA; mkgameaudio@gmail.com (M.K.); scott.t.barrows@jumpsimulation.org (S.B.); 4School of Medicine, Southern Illinois University, Springfield, IL 62702, USA; agoodson26@siumed.edu; 5College of Medicine, University of Illinois, Peoria, IL 61605, USA; 6School of Medicine, Southern Illinois University, Carbondale, IL 62901, USA; rbolinski98@siumed.edu; 7Mennonite College of Nursing, Illinois State University, Normal, IL 61790, USA; jrwille@ilstu.edu

**Keywords:** naloxone training, virtual reality, mixed reality, opioid overdose

## Abstract

**Background/Objectives:** The increased availability of naloxone underscores the urgent need for scalable, effective training interventions. While current training modalities show promise, critical challenges persist, particularly regarding the development of interactive, self-efficacious platforms that mitigate anxiety in real-world overdose response, especially among laypersons. Therefore, this study aimed to develop and evaluate the usability and acceptability of a novel, self-paced mixed reality-based training tool (ReviveXR). **Methods**: ReviveXR was designed using the Apple Vision Pro spatial computing headset and Unity platform, employing mixed reality technology to facilitate interaction with virtual overdose scenarios while maintaining awareness of the physical environment. The intervention included a simulated tutorial and interactive modules on overdose response, rescue breathing, and chest compressions. Field testing was conducted in two rounds across various settings with a heterogeneous sample (N = 25), including individuals who use drugs, bystanders, first responders, and technology specialists. Data collection involved pre- and post-intervention surveys and qualitative interviews. **Results**: Participants demonstrated significant improvements in knowledge related to overdose recognition, naloxone administration, rescue breathing, and chest compressions. ReviveXR increased participants’ confidence and intent to help overdose victims while reducing uncertainty during overdose reversal. Participants were predominantly from rural areas and primarily identified as White and male. Qualitative feedback emphasized the platform’s heightened engagement, realism, patient responsiveness, and capacity to enhance knowledge acquisition and behavioral preparedness compared with conventional training approaches. **Conclusions**: ReviveXR offers a scalable, cost-effective, engaging alternative to traditional naloxone training programs, demonstrating strong feasibility across diverse environments and participants. ReviveXR holds considerable promise for expanding and enhancing community overdose response capacities and training healthcare professionals and first responders.

## 1. Introduction

According to the Centers for Disease Control and Prevention (CDC), the age-adjusted rate of drug overdose deaths drastically increased from 8.2 per 100,000 in 2002 to 32.6 in 2022 [1]. Although the rate did not significantly change after 2021, drug overdose fatalities continue to remain at record levels, with an estimated 108,000 deaths in 2022 [1], which was higher than the number of deaths from motor vehicle accidents and gun violence combined. In other words, one American dies of a drug overdose nearly every five minutes around the clock. Most importantly, fatal drug overdoses are now the primary cause of death for Americans under 50 [2], surpassing all other leading causes. A recent national study indicated that almost one-third of Americans, equating to around 83 million adults, have experienced the death of someone they know due to a fatal drug overdose [3], and nearly one-fifth, or about 49 million adults, reported that the person they lost was a close friend or family member. There was no significant difference in overdose-related loss based on political party affiliation [3]. However, individuals who had lost someone to an overdose were significantly more inclined to consider addiction a critical or highly significant policy concern. For the same reason, combat against drug overdose has generally received bipartisan support; many recent policies and funding appropriation bills aimed at preventing drug overdoses, such as supporting addiction treatment programs and more effective harm reduction strategies, have found support from both major parties.

In 2020, an estimated 91,799 individuals died from opioid overdose, which was a 31% increase from 2019 [4]. Opioid overdose is the leading cause of the rising drug overdose deaths in the U.S. In 2000, opioids accounted for less than 50% of overdose fatalities; by 2021, that proportion had risen to more than 80%. The opioid overdose crisis began in the 1990s (wave 1) with a surge in deaths from prescription opioids, which was followed by a sharp increase in deaths related to heroin about a decade later (wave 2) [4]. Then, opioid-related death rates skyrocketed just before and during the COVID-19 pandemic (wave 3), fueled by the flooding of illicit fentanyl, a potent synthetic opioid that is approximately 50 times more potent than the semi-synthetic opioid heroin and 100 times more potent than the natural opioid morphine [4]. While fentanyl is approved by the FDA for use as an analgesic and anesthetic, illicit fentanyl is primarily smuggled through borders. Currently, in the U.S., 90% of deaths due to opioid overdose involve fentanyl and other synthetic opioids. In Illinois as well, overdose deaths due to synthetic opioids (e.g., fentanyl) increased by 3,341% in 2013–2022 (from 87 to 2,994), according to the Illinois Department of Public Health [5]. Furthermore, the opioid epidemic has permeated all strata of society over the last two decades, including all age groups (e.g., even newborns, due to maternal opioid use), income categories, and all counties across the nation [6]. The U.S. Congress Joint Economic Committee (JEC) released an analysis revealing that the opioid crisis costed the U.S. nearly USD 1.5 trillion in 2020, which marked a 37% increase from 2017 [7], the last time the CDC assessed the financial toll.

Several factors drive the emerging opioid epidemic among American youth. First, the youth opioid crisis is closely associated with the improper use of prescription opioids, characterized by taking medication in ways or doses not prescribed or without a doctor’s prescription. While opioid misuse and dependence are highest among young adults who are between 18 and 25 years old, a recent analysis of data from the Monitoring the Future study showed that 31% of high school seniors misuse prescription drugs [8]. Even legitimate use of prescription opioids before high school graduation was found to be independently associated with a 33% increase in the risk of future opioid misuse after high school [9]. A second, but related, problem is the diversion of prescription opioids. Youth are subject to pressure to share their prescription opioids for nonmedical use. Recent studies show that 14–35% of high school students divert their prescription opioids. Data has also shown that > 80% of youth receive diverted opioids as well [10]. The elevated rates of misuse are especially troubling, given that individuals who misuse prescription opioids are more prone to transitioning to heroin and fentanyl use [11]. Third, illegal drug use has become a serious problem among youth in recent years. Among teenagers, there was a staggering 169% increase in mortality associated with fentanyl and other illicit synthetic opioids from 2019 to 2020, followed by a 30% rise from 2020 to 2021 [1]. Fourth, the rate of overdose deaths linked to counterfeit pill use has more than doubled from 2019 to 2021 [12], because illicit fentanyl and other synthetic opioids are progressively contaminating the illegal drug supply. Those who succumbed to these counterfeit pills were generally younger and more likely to have a history of prescription medication misuse compared with those who died from other drug overdoses. These behaviors resulted in more than 6,000 deaths among youth (i.e., ages 15–24) in 2021 [13].

While educational and regulatory interventions for the prevention of opioid misuse and illegal opioid use are important, data suggest that harm reduction interventions for people who use drugs are urgently needed in a variety of settings. Therefore, as part of a broader array of solutions, harm reduction strategies have gained significant emphasis in recent years, with naloxone distribution at the forefront [14]. For example, 17% of opioid overdose deaths in Illinois had at least one nonfatal overdose in the prior 12 months, and 6% of those with a nonfatal overdose later died of an overdose [5], indicating a significant gap in harm reduction interventions such as the prevention of overdose fatalities among people who actively use drugs and those who are undergoing addiction treatment and recovery services. When breathing stops in an overdose, brain cells start dying after 5 min without oxygen [15]. However, the median time for an ambulance to arrive is 7 min in the U.S. It takes over 14 min in rural areas, and even 20 min or more in very rural locations [16], such as some southern counties in Illinois. As local emergency services do not have the capacity to provide necessary services in a timely manner, community empowerment initiatives play a crucial role in preventing overdose fatalities.

Therefore, in 2018, the U.S. Surgeon General issued a call to action for training laypersons who encounter people at risk for opioid overdose to administer naloxone nasal spray (i.e., brand name Narcan), the overdose reversal medication, which is now available in all states to anyone without a prescription and over the counter. The data suggests that in a vast majority of opioid overdose deaths, there was a potential bystander present, but unfortunately, naloxone was not administered [17]. While this can be due to the absence of naloxone at the incident, other common reasons are a lack of the necessary knowledge, skills, training, and confidence (i.e., self-efficacy) for naloxone administration. In addition, many people have negative attitudes toward people who use drugs and concerns regarding unintended consequences of attempting to help [18], such as being a suspect of involuntary manslaughter if the victim dies or the victim becoming aggressive post-recovery.

Accordingly, U.S. government agencies as well as many nonprofit and private sector organizations have implemented naloxone training programs for laypersons, but it is evident that the existing programs face major challenges in scalability, sustainability, securing community buy-in, addressing stigma, and improving efficacy in overdose management [18]. Most importantly, the development of training programs to improve laypersons’ self-efficacy in helping a victim and to reduce anxiety associated with attending a victim in real life as a good Samaritan remains a major challenge. Also, studies suggest that targeted naloxone distribution programs work best when naloxone is provided to people at risk of experiencing or witnessing an overdose [19], whereas comfort with carrying and administering naloxone was found to be crucial. Evidence has also shown that 80% of overdose reversals were carried out by individuals who also use drugs [20]. Hence, designing scalable and effective training programs, especially for vulnerable groups, that can improve opioid awareness and self-efficacy in overdose management remains a critical need.

In relation to life support training such as cardiopulmonary resuscitation (CPR), research has been conducted on the use of novel methods of training delivery that are both efficient and effective, specifically digital interventions such as virtual reality (VR). Particularly, VR interventions are more popular among youth due to their familiarity with technology and a strong affinity for immersive digital experiences, especially through gaming and social platforms. Although VR-based trainings are self-paced, they engage learners better than online trainings and make users feel more immersed in their surroundings compared with mannequin-based trainings [21,22]. However, their use in layperson naloxone training remains minimal [23]. To address this gap, the current study developed and tested the usability and acceptability of mixed reality-based naloxone training (ReviveXR), an immersive mixed reality (MR) prototype that combines digital imagery with the real world in front of a user’s eyes to recreate the experience of an actual situation of an overdose. This article is an expanded version of a paper that was presented at the 2024 annual meeting of the American Public Health Association [24]. The advantage of MR over VR is that it allows users to see and interact with virtual objects while simultaneously they are able to see their real-world surroundings [25]. This blended experience also avoids any feelings of isolation that can occur in a completely virtual environment and any anxiety that can occur in a completely physical environment. This study aimed to develop ReviveXR and test it with diverse participants and settings using an iterative process of programming and field testing to maximize its usability and acceptability.

## 2. Materials and Methods

### 2.1. Study Setting

The study was conducted in two higher education institutions in the Midwestern United States with funding from a collaborative innovation seed grant. Using an iterative process of development and testing, we developed the intervention and refined it based on feedback from the first round of implementation to ensure it was responsive to participant needs and usability. The MR prototype was developed in a VR laboratory setting in one university, and it was field tested in another university with community participants [24].

### 2.2. Study Design

Human subject components involved a single group pre/post design with two rounds of ReviveXR field testing that were three months apart—round 1 with initial prototype and round 2 with upgraded prototype—with first-round input from the participants. Design also included three points of data collection: baseline assessment (before round 1 training), round 1 post-training, and round 2 post-training.

### 2.3. ReviveXR Prototype Development

ReviveXR utilized cutting-edge spatial computing technologies, including the newly released Apple Vision Pro headset and real-time development tools such as the Unity game engine. The Apple Vision Pro is an advanced MR headset designed to seamlessly integrate digital content into the user’s real-world environment. Equipped with eye-tracking and gesture-based controls, it offers innovative ways to work, watch, and revisit memories, which provides an immersive and customizable training experience tailored to real-world scenarios. Unity is a versatile development platform and game engine that enables the creation of interactive 2D and 3D experiences, ranging from video games to simulations and virtual/augmented reality applications. Widely adopted by developers from diverse industries, it supports cross-platform development, making it a go-to tool for building immersive digital environments.

ReviveXR was developed in a way that it can merge virtual and physical worlds through MR technology. This allowed trainees to interact with virtual elements while staying aware of their real-world environment. The project involved tasks such as building an MR environment in Unity, selecting and implementing software development kits, designing the environment and audio, and refining the user interface and overall experience loop. The development followed a lean, vertical slicing approach to prioritize high-quality outcomes. A doctoral student contributed to the development of ReviveXR prototype, applying their expertise in areas such as 3D modeling, animation, programming, lighting design, and texture creation.

The training platform offered step-by-step training in overdose management, which includes CPR and the administration of intranasal naloxone, using CPR training guidelines provided by the American Heart Association (AHA) and naloxone training guidelines provided by the Substance Abuse and Mental Health Administration (SAMHSA). Trainees practiced using a real naloxone spray (i.e., demonstration device) on a virtual patient in their environment. Simultaneously, a virtual instructor in the background delivered an immersive tutorial in managing overdose situations. By combining MR with immersive video, ReviveXR created an interactive and realistic training experience, promoting effective skill acquisition in overdose response and CPR techniques.

ReviveXR participants learned the following skills: evaluation of an overdose victim’s surroundings for evidence of drug use and potential hazards that can be harmful to the responder (e.g., needles, fentanyl), patient’s responsiveness, contacting emergency services (i.e., calling 911), checking for breathing, assessing airways, performing CPR by administering chest compressions and rescue breathing, administering intranasal naloxone spray, reassessing the victim, and repeating previous steps based on reassessment. These features were upgraded and refined for usability in both community and laboratory environments based on feedback received in round 1 field testing.

### 2.4. Modifications of the Prototype

Based on feedback received from the round 1 field testing (see below), several improvements were made to enhance the MR experience in ReviveXR training. The initial introduction video was updated with an AI-generated person and voice to provide training guidance. New safety instructions were added to inform participants about protecting themselves from sharp objects like syringes and avoiding skin contamination from potent drugs such as fentanyl. Real-time instructions were integrated into key actions, including administering naloxone and performing rescue breathing, with an added sound notification to confirm successful completion of each step. The virtual phone used for calling 911 was removed, and the program was updated to detect hand movements simulating making the call. The mandatory rescue breathing cycles were reduced from 10 to 3 to shorten and streamline the process. The search for drug-related objects, such as syringes, was eliminated to keep the focus on life-saving actions. Improvements were made to the instruction panel as well, allowing users to reposition it more easily and align instructions with their actions. Additionally, bugs related to initial mannequin placement were resolved, ensuring a smoother training experience. The detailed changes made from round 1 to round 2 are available in Appendix A.

### 2.5. ReviveXR Field Testing

Field testing was conducted in two phases: rounds 1 and 2. Participants’ feedback from round 1 was utilized to modify the prototype for round 2. To evaluate applicability of MR training potential of ReviveXR across various real-world settings, field testing with participants was conducted in diverse environments such as offices, churches, living rooms of residences, backyards, and picnic areas. Human subjects data collection of the study received full-board approval from the Institutional Review Board (IRB) of the university that conducted the field testing. The training process, illustrated in Figure 1, included steps such as calibrating the Apple Vision Pro headset, selecting the ReviveXR app, following VR instructions, and engaging in simulated overdose response activities, including naloxone administration and rescue breathing to successfully achieve patient recovery. A short ReviveXR demo video can be found in the QR code of Figure 1 and this link [26].

Participants (N = 25) were recruited from both urban and rural communities as well as various organizations through investigator-initiated contacts, with incentives provided as a USD 25 gift card upon completion of each round of data collection. The study recruited a diverse sample representing people who use drugs, their family members, emergency room physicians, registered nurses, first responders (i.e., police officers, ambulance workers, and firefighters), IT and VR professionals, university professors and students, and health program coordinators who regularly conduct overdose education and naloxone distribution (OEND) in communities. To reduce the dropout rate, participants who were unsure about their availability for participation in the second round of field testing approximately three months after the first round were excluded from the first round as well.

Baseline assessment, conducted through a survey prior to round 1 training, included participants’ demographics, experience encountering an overdose event (i.e., self or others), opioid-related knowledge [27], prior naloxone trainings received, and comfort levels regarding naloxone use and emergency response procedures such as CPR [28]. Two more surveys were administered after round 1 training and round 2 training, respectively, which were conducted approximately three months apart using the same post-training questionnaire. In addition to post-training knowledge, skills, and confidence related to overdose response, these surveys evaluated participants’ learning self-efficacy using a modified version of the Learning Self-Efficacy Scale (L-SES) adjusted for naloxone [29]. After responding to quantitative paper-and-pencil surveys in both round 1 and round 2, participants shared their training experiences and perspectives through semi-structured interviews, providing more in-depth complementary qualitative feedback.

### 2.6. Statistical Methods

This study involved pre- and post-training assessments, including written surveys and qualitative interviews, to evaluate changes in participants’ knowledge, comfort levels, and self-efficacy regarding naloxone administration and emergency response protocols (e.g., CPR). Data were collected in two rounds: the first round focused on an initial simulation, and the second round assessed the impact of an improved version of the VR headset program, which was updated based on user feedback after the first round.

To assess changes in knowledge and skill acquisition, descriptive statistics were computed, and paired *t*-tests were conducted to compare pre- and post-training scores using SPSS Version 29 [30]. Prior to analysis, normality of the data was tested, and any missing values were addressed using complete case analysis. A significance level of 5% was set for all statistical tests.

Results from round 1 and round 2 post-training questionnaires were compared to evaluate the impact of the program enhancements. The same statistical approach was applied, including paired *t*-tests and descriptive statistics, with normality checks and complete case analysis.

Qualitative data from semi-structured interviews were examined using contextual analysis to obtain a deeper understanding of ReviveXR training experience in the sociocultural, professional, and situational context of participants along with their suggestions for further improvements. The analysis aimed to understand the usability and acceptability of ReviveXR from different perspectives and reveal insights that are not immediately apparent when using qualitative analyses.

## 3. Results

Of the 25 participants recruited in round 1, most (76%) lived in rural areas, with smaller percentages in suburban (12%) and urban (12%) settings. The gender distribution showed that 64% were men, 28% were women, and 8% identified as other. In terms of race, 64% were White or Caucasian, 24% were Asian or Pacific Islander, and 4% were either Black, bi-racial, or multi-racial. Also, most respondents (96%) were non-Hispanic. Education level was relatively high, with 28% each holding a bachelor’s, master’s, or doctoral degree; only 16% had less than a graduate degree. Twenty of the twenty-five participated in round 2 training as well (80% follow-up rate); three of the five people who were unable to participate in round 2 training three months later had moved out of the area, and two were unavailable due to other commitments.

When asked about the experience using VR or MR and encountering overdose, nearly half (48%) reported that they had used VR or MR previously, while 36% had witnessed an overdose. Notably, no one had personally experienced an overdose. In training, 52% reported that they had been trained in administering naloxone spray, 72% in rescue breathing, and 76% in chest compressions, but only 20% had intervened in an overdose using naloxone.

### 3.1. Effects on Knowledge

From baseline to round 1 post-training, participants demonstrated significant improvements in knowledge (Table 1). Correctly identifying slow/shallow breathing as an overdose sign significantly increased from 80% to 100% (*p* = 0.025) and very small pupils from 52% to 88% (*p* = 0.003), whereas incorrectly identifying fits (convulsions) as an overdose sign significantly decreased from 24% to 8% (*p* = 0.046). Additionally, participants’ understanding of the duration of naloxone’s effect increased from 32% to 72% (*p* = 0.002), whereas the knowledge of the need for a second naloxone dose improved from 72% to 100% (*p* = 0.008). Correctly identifying loss of consciousness as an opioid overdose sign increased from 96% to 100%, and incorrectly identifying rapid heartbeat decreased from 20% to 12%, while participants’ ability to correctly select the time duration for naloxone to start having effect increased from 76% to 96%, but these improvements were statistically not significant. In round 2 post training, knowledge measures did not show any improvement, and participants’ awareness of the duration of naloxone’s effect and their ability to correctly select the time duration for naloxone to start having an effect decreased significantly (from 72% to 40% and from 96% to 80%, respectively), compared with round 1 (Table 2; see Section 4).

In terms of knowledge on managing an opioid overdose as measured in 5-point Likert scales, perceived effectiveness of rescue breathing increased from a mean of 3.84 at baseline to 4.68 at round 1 post-training (*p* < 0.001), and confidence in having enough information to effectively respond to an overdose rose from 3.36 to 4.76 (*p* < 0.001) (Table 1). Two other measures showed some improvement but were not statistically significant: perceived effectiveness of chest compressions (from 3.44 to 3.72) and perceived effectiveness of naloxone (from 4.84 to 5.00). Mean for incorrectly answering the question, “There is no need to call for an ambulance if I know how to manage an overdose” was already very low at baseline (0.04), and did not show any further reduction in round 1 post-training but decreased further to zero in round 2 post-training.

### 3.2. Effects on Confidence

Confidence, which was scaled from 1 to 5, showed significant improvements after round 1 training (Table 1). Confidence in performing rescue breathing increased from a mean of 3.60 to 4.60 (*p* < 0.001), and confidence in administering naloxone rose from 3.60 to 4.88 (*p* < 0.001). Although confidence in performing chest compressions also improved, the change was not statistically significant (*p* = 0.083). Most importantly, participants’ comfort level in responding to an overdose situation significantly increased, with a mean change from 3.44 to 4.44 (*p* = 0.001). In round 2 post-training, confidence in administering rescue breathing, chest compressions, and naloxone decreased slightly, and comfort level in responding to an overdose situation also decreased (Table 2; see Section 4).

### 3.3. Effects on Uncertainty

Uncertainty, which was measured on a scale from 1 (strongly disagree) to 5 (strongly agree), showed significant improvements after round 1 training (Table 1). Participants’ fear of making mistakes in overdose situations dropped significantly from a mean of 2.68 to 1.96 (*p* = 0.023). The need for additional naloxone training also significantly decreased, with the mean shifting from 2.80 to 1.88 (*p* = 0.001). The need for further rescue breathing training showed some improvement (from 2.64 to 2.08), which was not statistically significant (*p* = 0.055). Unlike knowledge and confidence indicators, uncertainty further decreased overall with the additional training in round 2 (Table 2). Particularly, fear of making mistakes in overdose situations and the need for more rescue breathing training decreased from round 1 to round 2, although the change was not significant.

### 3.4. Effects on Intent

Round 1 post-training showed significant improvements in participants’ intent to intervene in overdose situations, with all intent measures increased in a scale from 1 (No intent) to 5 (Very High): intent to use naloxone from a mean of 4.16 to 4.84 (*p* = 0.005), intent to use rescue breathing from 4.12 to 4.64 (*p* = 0.006), and intent to respond to an overdose from 4.28 to 4.88 (*p* = 0.01) (Table 1). Although the intent to perform chest compressions showed some improvement (from 3.88 to 4.32), this change was not statistically significant (*p* = 0.069). Similar to the knowledge and confidence indicators in round 1, intent indicators decreased in an undesirable direction after round 2 training (Table 2; see Section 4).

### 3.5. Learning Self-Efficacy

The Learning Self-Efficacy Scale (L-SES) was used for measuring learning self-efficacy in naloxone administration. According to the round 1 post-training evaluation, all indicators in all aspects of learning self-efficacy (i.e., cognitive, affective, and psychomotor) improved significantly (Table 3). In round 2 post-training, 9 of the 12 indicators (i.e., all cognitive, 1 of the 4 affective, and all psychomotor) improved significantly (Table 4).

### 3.6. Qualitative Feedback

Qualitative feedback was positive overall, with valuable suggestions for improvement (Table 5). Regarding the physical fit and comfort level of the headset, participants across both testing rounds reported mixed experiences. While many found it to be manageable and easy to adjust for the duration of the ReviveXR, some common complaints included headset heaviness, eye strain and dryness, and compatibility issues with eyeglasses. As for immersion and realism, feedback varied considerably, though round 2 showed improved graphics quality with more realistic patient appearance. Across both rounds, participants appreciated the haptic feedback that triggered when they performed correctly. A suggestion worth noting was to add ambient sounds or have the training set in a real environment (e.g., a grocery store) to enhance the level of immersion. Regarding engagement and affect, the training content was generally well-received as easy to follow and not redundant. Users appreciated the ability to move around and have hands-on experiences, which kept the training engaging. Participants typically reported feeling fine and good in both rounds, suggesting the experience was emotionally neutral and comfortable.

When comparing ReviveXR to traditional overdose-reversal training methods (many of them utilizing either instructional videos or live instructors with plastic mannequins), ReviveXR demonstrated unique advantages. These benefits included heightened engagement, efficiency for group training, and the ability to train muscle memory to enhance confidence. The content was considered comprehensive and succinct as well, though ReviveXR lacks the ability to provide users with detailed feedback on their performed actions (something that traditional methods account for). Overall, participants reported improvements in round 2, specifically with interaction. In round 1, struggles with virtual object manipulation detracted from the educational experience. And though some technical challenges remained in round 2 (e.g., confusion with the Panel Movement button), the removal of the object scanning section and general improved graphics were widely appreciated and contributed to a smoother experience for almost all participants.

## 4. Discussion

ReviveXR is a self-paced naloxone training that uses MR to allow trainees to interact with virtual objects while seeing their surroundings. The key innovation in this research is the use of MR to create an experience that is very close to an actual opioid overdose situation. It involves simulated tutorials and interactive overdose response activities, rescue breathing, and chest compression using the Apple Vision Pro spatial computing headset. It was field tested in diverse environments with diverse participants whose feedback was used to modify the prototype; the revised prototype was field tested again. Participants’ overdose-related knowledge, confidence, and intent to intervene in overdose situations significantly increased after ReviveXR, and uncertainty in overdose management significantly decreased; also, the learning self-efficacy was significant.

Improvements in overdose-related knowledge in this research are compatible with prior findings that naloxone training leads to substantial improvements in knowledge, particularly in recognizing overdose signs and understanding naloxone’s effects [31]. Previous studies also reported similar increases in knowledge, especially concerning the duration of naloxone and its use for multiple doses [32]. However, as with past research, there remains a gap in real-world application, suggesting that knowledge improvements may not always translate into confident intervention [28,33]. This highlights the need for more practical training components to complement knowledge-based instruction. Participants also reported increased perceived effectiveness in overdose response after training, similar to previous studies, which showed that participants feel more capable to intervene in overdose situations, particularly in using naloxone and performing rescue breathing [28,33]. The consistent decrease in the misconception about the need to call an ambulance further highlights the importance of reinforcing this message in naloxone education programs to ensure a comprehensive understanding of overdose management [34,35].

Additionally, the improvements in confidence observed in this study are supported by prior research that consistently shows that such training boosts confidence among laypersons in responding to opioid overdoses [36]. Previous research has also highlighted the importance of comfort levels in overdose response, with increased comfort often correlating with a greater likelihood of intervention in real-life situations [37]. While confidence in chest compressions improved in this study, the lack of statistical significance, coupled with uncertainty about the ability to conduct rescue breathing, mirrors prior research. This finding suggests that, while participants feel more confident, certain skills (like chest compressions and rescue breathing) may require additional practice or instruction to ensure effective intervention [38].

As one would expect, increased confidence among our sample was coupled with decreased uncertainty. The current literature suggests that fear or uncertainty often prevents laypersons from using their training in real-life situations [28,33]. In this study, we saw significant decreases in levels of uncertainty among our participants. Certainly, our findings support this notion, as intent to act in overdose situations increased following mixed reality-based training.

Previous studies have shown that the learning self-efficacy, particularly in the cognitive and psychomotor domains, is important in developing clinical skills [29]. Affective self-efficacy, or emotional readiness, is also crucial, as it impacts the participant’s willingness to act despite potential stress or uncertainty [28,39]. These results suggest that comprehensive training that addresses both the knowledge and practical application of naloxone use significantly enhances participants’ self-efficacy [39,40]. However, continued research is needed to explore how these improvements translate into real-world overdose interventions.

Finally, round 2 post-training assessment revealed a decline in opioid-related knowledge, confidence in overdose management, and intent to intervene in overdose situations compared with round 1 post-training assessment, although these differences were not statistically significant. The slight change in outcomes in the undesirable direction in round 2 is potentially attributed to the replacement of the human instructor with an AI-generated person and voice to provide training guidance.

Contrarily, intent to help an overdose victim further improved from round 1 to round 2, probably because of the language added to round 2 to address potential questions trainees might have regarding revival of overdose victims, such as being ready to respond to reactions of the recovered patient. The final version of ReviveXR that will be used in the beta testing prior to a community-based randomized controlled trial will consist of positive features from both rounds.

### 4.1. Implications for Future Practice and Research

This is the first study to use MR technology to blend the virtual and physical environments. This way, people can visualize and interact with virtual objects, but simultaneously, they can also see their real-world surroundings. This blended experience aims to avoid feelings of isolation that can occur in a completely virtual environment and avoid feelings of anxiety that can occur in a completely physical environment. Additionally, training participants are able to use real-world objects in a virtual environment, for example, administering a real naloxone dose to a virtual patient. This approach will increase the likelihood that users can draw upon their memory from the experience if they must save a real victim, regardless of whether it is on a street in a big city or on a farm field in a small town [24]. For example, big cities have naloxone hotspots, and ReviveXR can bring that environment to the person so that they feel familiar with it. Then, participants can draw upon their memory if they must save a real victim. The assumption is that, after people go through this simulated immersive environment that will teach how to use naloxone, they will feel more confidence and less panic when they help an actual overdose victim.

According to evidence from empirical naloxone studies available in electronic databases [41,42,43,44,45,46], technology-based, self-paced interventions such as ReviveXR can be conveniently administered in institutional settings, e.g., prisons, hospitals, and universities, as well as community-based programs, e.g., mobile clinics and nursing outreach services. ReviveXR can be easily administered in a variety of settings, utilizing minimal human and physical resources. Also, Wi-Fi is not required once the program is downloaded on the device. In the current version of ReviveXR, any person who can be trained for operation of a VR headset will be able to administer the training, even if they do not have certification as a trainer of basic life support. Instead of receiving one-on-one instructions from an instructor, trainees follow virtual instructions and navigate through the modules in a self-paced manner, i.e., at their desired pace. Moreover, the optimization of resource utilization increases scalability and sustainability.

Another advantage of ReviveXR is the ability to upgrade the user experience based on advancements in technology and new policies, guidelines, and laws. For example, in future upgrades, the ReviveXR app can combine several interactive technologies and educational materials such as video animation of what is happening inside the body of an overdose victim. The field of MR is advancing rapidly, new laws can emerge, and clinical guidelines may change, for example, newer versions of the headset, more effective medications and CPR guidelines, and changes in Good Samaritan Laws. Likewise, ReviveXR can be easily upgraded remotely to enhance the user’s experience and provide more up-to-date information. Similarly, the opioid epidemic is emerging among youth, and they are tech-savvy, so examining what works best for youth will be helpful.

Finally, a logical next step for ReviveXR is a randomized controlled trial (RCT) that can examine its effectiveness and sustainability compared with widespread in-person naloxone training methods in communities. The RCT should also follow up with participants over years to examine how they use the training to save lives in real-world settings, if/when they need refresher training, as well as logistical issues they might encounter, such as obtaining a resupply of naloxone. Obviously, the RCT should have high internal and external validity in order to interpret broader impacts.

### 4.2. Limitations

Several limitations should be acknowledged in the current implementation and evaluation of ReviveXR. First, variability in digital and technical literacy among users may pose a barrier to effective engagement with the MR platform. Second, the long-term retention of skills acquired through ReviveXR remains uncertain, particularly for individuals who may encounter overdose events infrequently. Third, the realistic and immersive nature of overdose scenarios may be distressing for some participants, especially those with personal or traumatic experiences related to opioid use and overdose, raising ethical considerations around potential re-traumatization. Lastly, the current evidence is insufficient (modest sample size) to determine whether ReviveXR is equivalent or superior to conventional in-person naloxone training methods in improving real-world outcomes. Further research, including a community-based randomized controlled trial, is necessary to evaluate its effectiveness across diverse populations, including variations in age, education, and skill status. Addressing these limitations will also require thoughtful program design, oversampling of vulnerable groups, and sustained research efforts to ensure ReviveXR is both effective and inclusive.

## 5. Conclusions

ReviveXR proved feasible across various settings and participant groups. The findings demonstrated increased knowledge, confidence, willingness to intervene, and reduced hesitation around naloxone use. By incorporating MR into a flexible, self-directed training format, ReviveXR not only boosted engagement but also delivered real-time feedback, presenting an engaging approach to improving naloxone skills among laypersons and first responders alike. ReviveXR also offers a scalable, cost-effective alternative to traditional naloxone training programs, demonstrating strong feasibility across diverse environments and participant backgrounds. While the initial findings support its usability and acceptability, future research should explore the long-term impact of the training, including how user familiarity influences sustained confidence, knowledge retention, intent to act, and behavioral readiness, and it should consider comparing these outcomes with those of participants receiving refresher training at set intervals. Additionally, participant diversity—such as prior experience with technology or overdose response—may shape short- and long-term usability outcomes and should be further examined. These findings may be contextualized within the existing literature on the extended use of immersive training tools, which show promise in enhancing procedural memory and self-efficacy over time [47,48]. Such research can inform refinement, scalability, and equitable deployment of ReviveXR across varied populations and settings. Additionally, the physical space required for the current training modality may limit scalability, particularly in rural or resource-limited settings. Thus, further development of more compact, mobile-friendly prototypes is needed. Lastly, since the current approach allows for only one participant at a time without multiple devices—raising cost and efficiency concerns—future work should aim to refine the prototype and assess its effectiveness to support potential funding efforts for broader implementation.

## Figures and Tables

**Figure 1 healthcare-13-01449-f001:**
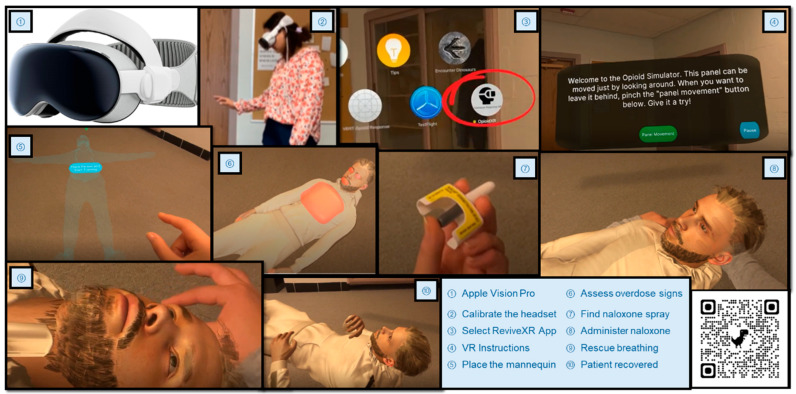
Illustration of user experience in mixed reality naloxone training (ReviveXR) with the VR headset.

**Table 1 healthcare-13-01449-t001:** Baseline and round 1 training comparison of overdose-related knowledge, confidence, uncertainty, and intention to intervene in opioid overdose situations.

Survey Variable	Baseline Mean or Proportion (SD/SE)	Round 1 Mean or Proportion (SD/SE)	z or t Score	*p* Value
**Overdose-Related Knowledge**				
Slow/shallow breathing (Correct = Yes) ^a^	0.80 (0.09)	1.00 (0.00)	2.24	0.025
Fits or convulsions (Correct = No) ^a^	0.24 (0.17)	0.08 (0.19)	−2.00	0.046
Rapid heartbeat (Correct = No) ^a^	0.20 (0.18)	0.12 (0.19)	−1.00	0.317
Loss of consciousness (Correct = Yes) ^a^	0.96 (0.04)	1.00 (0.00)	1.00	0.317
Very small pupils (Correct = Yes) ^a^	0.52 (0.14)	0.88 (0.07)	3.00	0.003
None of the above (Correct = No) ^a^	0.00 (0.00)	0.00 (0.00)	-	-
How long does naloxone take to start having effect? (2–5 min) ^b^	0.76 (0.10)	0.96 (0.04)	1.89	0.059
How long do the effects of naloxone last for? (1 h) ^b^	0.32 (0.16)	0.72 (0.11)	3.16	0.002
If the 1st dose of naloxone has no effect, a 2nd can be given. (True) ^a^	0.72 (0.11)	1.00 (0.00)	2.65	0.008
No need to call for an ambulance if I can manage overdose. (False) ^a^	0.04 (0.20)	0.04 (0.20)	-	-
Rescue breathing is effective when managing an opioid overdose: (1–5) ^c^	3.84 (1.03)	4.68 (0.63)	4.68	0.000
Chest compressions are effective when managing an opioid overdose: (1–5) ^c^	3.44 (1.04)	3.72 (1.49)	0.98	0.337
Naloxone is effective when managing an opioid overdose: (1–5) ^c^	4.84 (0.47)	5.00 (0.00)	1.69	0.103
I have enough information to respond to an overdose effectively. (1–5) ^c^	3.36 (1.55)	4.76 (0.44)	4.85	0.000
**Confidence in Managing an Overdose**				
How confident are you in performing rescue breathing? (1–5) ^c^	3.60 (1.55)	4.60 (0.58)	4.08	0.000
How confident are you in performing chest compressions? (1–5) ^c^	3.44 (1.71)	4.04 (1.46)	1.81	0.083
How confident are you in performing naloxone administration? (1–5) ^c^	3.60 (1.47)	4.88 (0.33)	4.57	0.000
How comfortable do you feel responding to an overdose? (1–5) ^c^	3.44 (1.45)	4.44 (0.82)	3.87	0.001
**Uncertainty in Managing an Overdose**				
I would be afraid of doing something wrong in an overdose situation (1–5) ^c^	2.68 (1.60)	1.96 (1.34)	−2.42	0.023
I would need more naloxone training (1–5) ^c^	2.80 (1.66)	1.88 (1.13)	−3.87	0.001
I would need more rescue breathing training (1–5) ^c^	2.64 (1.68)	2.08 (1.50)	−2.02	0.055
**Intent to Intervene in an Overdose Situation**				
Intent to utilize naloxone nasal spray (1–5) ^c^	4.16 (1.25)	4.84 (0.47)	3.07	0.005
Intent to utilize rescue breathing (1–5) ^c^	4.12 (0.97)	4.64 (0.81)	2.98	0.006
Intent to respond to an overdose event when notified (1–5) ^c^	4.28 (1.17)	4.88 (0.33)	2.78	0.010
Intent to utilize chest compressions (1–5) ^c^	3.88 (1.20)	4.32 (1.25)	1.90	0.069

^a^ Dichotomous response, hence used the proportion test. ^b^ Multiple-choice questions, hence used the proportion test. ^c^ Likert-scale questions with higher scores indicating more desirable response.

**Table 2 healthcare-13-01449-t002:** Round 1 and round 2 training comparison of overdose-related knowledge, confidence, uncertainty, and intention to intervene in opioid overdose situations.

Survey Variable	Round 1 Mean or Proportion (SD/SE)	Round 2 Mean or Proportion (SD/SE)	z or t Score	*p* Value
**Overdose-Related Knowledge**				
Slow/shallow breathing (Correct = Yes) ^a^	1.00 (0.00)	1.00 (0.00)	-	-
Fits or convulsions (Correct = No) ^a^	0.10 (0.21)	0.10 (0.21)	0.00	1.000
Rapid heartbeat (Correct = No) ^a^	0.15 (0.21)	0.05 (0.22)	−1.00	0.317
Loss of consciousness (Correct = Yes) ^a^	1.00 (0.00)	1.00 (0.00)	-	-
Very small pupils (Correct = Yes) ^a^	0.85 (0.09)	0.80 (0.10)	−0.45	0.655
None of the above (Correct = No) ^a^	0.00 (0.00)	0.00 (0.00)	-	-
How long does naloxone take to start having effect? (2–5 min) ^b^	0.96 (0.04)	0.80 (0.09)	−2.00	0.046
How long do the effects of naloxone last for? (1 h) ^b^	0.72 (0.11)	0.40 (0.15)	−2.31	0.021
If the 1st dose of naloxone has no effect a 2nd can be given. (True) ^a^	1.00 (0.00)	1.00 (0.00)	-	-
No need to call for an ambulance if I can manage overdose. (False) ^a^	0.05 (0.22)	0.00 (0.00)	−1.00	0.317
Rescue breathing is effective when managing an opioid overdose: (1–5) ^c^	4.80 (0.52)	4.80 (0.41)	0.00	1.000
Chest compressions are effective when managing an opioid overdose: (1–5) ^c^	3.60 (1.60)	3.05 (1.61)	−1.93	0.069
Naloxone is effective when managing an opioid overdose: (1–5) ^c^	5.00 (0.00)	5.00 (0.00)	-	-
I have enough information to respond to an overdose effectively. (1–5) ^c^	4.75 (0.44)	4.75 (0.44)	0.00	1.000
**Confidence in Managing an Overdose**				
How confident are you in performing rescue breathing? (1–5) ^c^	4.60 (0.60)	4.45 (0.83)	−1.14	0.267
How confident are you in performing chest compressions? (1–5) ^c^	3.90 (1.59)	3.70 (1.49)	−0.94	0.359
How confident are you in performing naloxone administration? (1–5) ^c^	4.90 (0.31)	4.85 (0.49)	−0.57	0.577
How comfortable do you feel responding to an overdose? (1–5) ^c^	4.45 (0.83)	4.25 (1.25)	−0.70	0.494
**Uncertainty in Managing an Overdose**				
I would be afraid of doing something wrong in an overdose situation (1–5) ^c^	2.00 (1.38)	1.70 (0.98)	−1.00	0.330
I would need more naloxone training (1–5) ^c^	1.85 (1.14)	1.90 (1.21)	0.19	0.853
I would need more rescue breathing training (1–5) ^c^	2.20 (1.54)	1.70 (1.08)	−1.39	0.180
**Intent to Intervene in an Overdose Situation**				
Intent to utilize naloxone nasal spray (1–5) ^c^	4.80 (0.52)	4.65 (0.67)	−1.83	0.083
Intent to utilize rescue breathing (1–5) ^c^	4.60 (0.88)	4.25 (1.16)	−2.10	0.049
Intent to respond to an overdose event when notified (1–5) ^c^	4.85 (0.37)	4.55 (0.89)	−2.04	0.055
Intent to utilize chest compressions (1–5) ^c^	4.15 (1.35)	4.10 (1.33)	−0.29	0.772

^a^ Dichotomous response, hence used the proportion test. ^b^ Multiple-choice questions, hence used the proportion test. ^c^ Likert-scale questions with higher scores indicating more desirable response.

**Table 3 healthcare-13-01449-t003:** The Learning Self-Efficacy Scale (L-SES) for naloxone administration in round 1 post-training assessment.

Survey Variable	Round 1 Mean (SD)	t Score	*p* Value
**Cognitive**			
I can recall how to perform naloxone administration. (1–5) ^a^	4.68 (0.85)	9.85	0.000
I understand the content of naloxone administration and can demonstrate it to others. (1–5) ^a^	4.64 (0.57)	14.42	0.000
I can verbally explain the purpose and principle of operating naloxone administration. (1–5) ^a^	4.76 (0.44)	20.19	0.000
I can verbally explain the sequence and interrelationship between each step. (1–5) ^a^	4.64 (0.64)	12.86	0.000
**Affective**			
I think it would be more beneficial if there was more of a focus on the naloxone training than the background information. (1–5) ^a^	3.00 (1.19)	0.00	1.000
I think I gain more in learning about naloxone administration than in other content. (1–5) ^a^	3.88 (1.13)	3.89	0.001
I tend to pay more attention to information related to naloxone administration content. (1–5) ^a^	3.88 (1.17)	3.77	0.001
I tend to actively look for information related to naloxone administration. (1–5) ^a^	3.84 (1.21)	3.46	0.002
**Psychomotor**			
I can precisely imitate the instructor’s steps and actions of naloxone administration. (1–5) ^a^	4.28 (0.89)	7.19	0.000
I can smoothly complete the operation steps of naloxone administration. (1–5) ^a^	4.56 (0.58)	13.38	0.000
I try to monitor my naloxone administration for improvements. (1–5) ^a^	4.52 (0.71)	10.64	0.000
I try to monitor my naloxone administration operations and make proper adjustments as needed. (1–5) ^a^	4.48 (0.77)	9.61	0.000

^a^ Likert-scale questions with higher scores indicating more desirable response.

**Table 4 healthcare-13-01449-t004:** The Learning Self-Efficacy Scale (L-SES) for naloxone administration in round 2 post-training assessment.

Survey Variable	Round 2 Mean (SD)	t Score	*p* Value
**Cognitive**			
I can recall how to perform naloxone administration. (1–5) ^a^	4.80 (0.41)	19.62	0.000
I understand the content of naloxone administration and can demonstrate it to others. (1–5) ^a^	4.70 (0.57)	13.31	0.000
I can verbally explain the purpose and principle of operating naloxone administration. (1–5) ^a^	4.70 (0.47)	16.17	0.000
I can verbally explain the sequence and interrelationship between each step. (1–5) ^a^	4.65 (0.59)	12.57	0.000
**Affective**			
I think it would be more beneficial if there was more of a focus on the naloxone training than the background information. (1–5) ^a^	3.00 (1.08)	0.000	1.000
I think I gain more in learning about naloxone administration than in other content. (1–5) ^a^	3.35 (1.18)	1.324	0.201
I tend to pay more attention to information related to naloxone administration content. (1–5) ^a^	3.55 (1.10)	2.238	0.037
I tend to actively look for information related to naloxone administration. (1–5) ^a^	3.40 (1.10)	1.633	0.119
**Psychomotor**			
I can precisely imitate the instructor’s steps and actions of naloxone administration. (1–5) ^a^	4.50 (0.69)	9.75	0.000
I can smoothly complete the operation steps of naloxone administration. (1–5) ^a^	4.65 (0.49)	15.08	0.000
I try to monitor my naloxone administration for improvements. (1–5) ^a^	4.00 (0.97)	4.60	0.000
I try to monitor my naloxone administration operations and make proper adjustments as needed. (1–5) ^a^	4.30 (0.86)	6.73	0.000

^a^ Likert-scale questions with higher scores indicating more desirable response.

**Table 5 healthcare-13-01449-t005:** Qualitative feedback in round 1 and round 2 post-training assessments.

Parameter	Round 1 Training	Round 2 Training
Fit/Comfort	Some reported physical discomfort during use, including headset heaviness, mild headaches after extended use, eye strain, and difficulty fitting the headset with glasses. Positive feedback included comfort and ease of use, with comments like “didn’t notice I had headset on,” “manageable and comfortable,” and “easy to adjust.” Suggestions included a better nose-piece and a more comfortable headband.	Most participants found the headset adjustable and its weight manageable for shorter ReviveXR sessions (~15 min). However, some reported issues with eye strain, headset weight/sliding, and compatibility with glasses.
Immersion	Participants had mixed feedback regarding the realism of the simulation. Some felt aware of the simulation, describing it as “looking like a videogame,” while others found it realistic, with one expressing surprise when the patient responded to their actions. Concerns about realism included unnatural patient behavior and unrealistic procedures. A suggestion was adding ambient sounds (e.g., store noise).	Participants mentioned the patient looking realistic, though some commented that not having a tactile response with the patient contributed to lower immersion level Participants noted that the haptic responses provided by the headset (triggered by performing actions correctly) enhanced immersion experience
Engagement	Participants generally had positive feedback, expressing engagement and noting the clarity and simplicity of ReviveXR, with comments such as “easy” and “straightforward.” However, there was an instructional timing issue, with one participant mentioning that it was challenging when the instructor spoke while they were still processing actions, describing it as “incongruent.”	Most participants found the content and VR experience engaging, interesting, and appropriately concise, with feedback such as “not redundant, easy to follow.” The ability to move around and have hands-on experience was highlighted as a key factor in maintaining engagement throughout the training.
Affect	No serious emotional responses reported. Felt “fine” was a common response.	Participants overall felt “good” and “unstressed”.
Interactivity	Challenges were awkward interactions with virtual objects, uncertainty about action recognition, and difficulty adjusting to the pinch motion. Navigation was easy, and the pass-through component was praised.	Participants stated that moving through and interacting with ReviveXR was intuitive.
Comparison with the traditional training	The use of patient response simulations was praised for enhancing realism and improving efficiency compared with traditional CPR training with plastic busts. The approach was seen as an effective method for building knowledge and confidence in assisting with opioid overdoses. However, some noted that the focus on technology occasionally detracted from the core training content.	The incorporation of VR-based approaches received a positive reception. Participants highlighted the realism of the experience, with one noting that it closely resembles real-world scenarios and aids in developing muscle memory. However, the absence of tactile feedback, which is typically provided by physical mannequins, was noted as a limitation. One pointed out that while ReviveXR is effective for general use, it may not be as well-suited for techniques requiring precise hands-on skills, such as CPR.
Content improvements needed	The rescue breathing portion was seen as overly emphasized, with some participants feeling it received disproportionate focus. Also, the activity requiring participants to identify objects, including paraphernalia, was considered unnecessary by some.	A participant felt that it needs to be noted that rescue breaths require a barrier/mask. Overall, participants considered content to be comprehensive and succinct.
Technical improvements needed	Participants noted occasional glitches with object placement and instructional text visibility. Incorporating introductory video into main training platform was suggested.	Improved graphics and removal of object scanning were considered by participants as significant improvements and generally noted a smoother experience. Participants reported initial confusion with the Panel Movement button and difficulty seeing the chest rise and fall despite clear instructions. Additionally, the phone interaction was unclear, with one participant mentioning uncertainty about when to pull the phone out of their pocket during the training.

## Data Availability

The data presented in this study are available in de-identified form on request from the corresponding author due to privacy and ethical reasons.

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
