# Peer review of "Usability of Mixed Reality for Naloxone Training: Iterative Development and Field Testing of ReviveXR"

_healthcare, 2025, doi:10.3390/healthcare13121449_

Round 1
Reviewer 1 Report
Comments and Suggestions for Authors
The manuscript developed and evaluated the usability and acceptability of Virtual Reality Naloxone Training (VeNT), an immersive mixed reality (MR) prototype that combines digital visuals with the user’s real-world surroundings to simulate a realistic opioid overdose scenario. This manuscript is an extended version of a paper previously presented at an annual conference, with the key enhancement being the incorporation of mixed reality (MR). MR offers advantages over traditional virtual reality (VR) by allowing users to interact with virtual objects while still perceiving their real-world environment.
Below are several suggestions to improve the manuscript:
-
While the application is named Virtual Reality Naloxone Training (VeNT), the extended version submitted for publication emphasizes mixed reality. Please clearly distinguish the original VR-based work from this new MR-based version, particularly in the Introduction and Conclusions sections. This distinction is critical to avoid confusion and to highlight the added value of MR.
-
Section 2.1. Study Setting and Section 2.2. Study Design sections should be improved. What is the implementation mechanism of the iterative development process? What factors contributed to the differences between the two prototype versions in the rounds? While Section 2.3. Modifications of the Protoype addresses some of these differences, a comparison table summarizing key modifications between rounds would greatly assist reader comprehension.
-
A more explicit comparison between VeNT and current standard naloxone training methods regarding efficacy and user experience would help clarify the benefits of the VeNT. Again, a comparison table could help summarize these points effectively.
-
The statement, "The analysis aimed to understand the usability and acceptability of VeNT from different perspectives and reveal insights that are not immediately apparent when using quantitative analyses. " suggests that usability and acceptability are the core constructs in the study. Are there any specific, validated quantitative metrics to assess these dimensions?
- For sample size & generalizability, is there any way to find an approximate sample size using an statistical analysis technique such as G-power analysis? Because the number of participants seem a bit low for the questions that must be answered in Likert scale and binary questions. Considering the diversity of participants (e.g., individuals who use drugs, bystanders, first responders, and technology specialists), a more thorough discussion of sample adequacy is warranted. Including such an analysis would help justify the generalizability of your findings.
- The Conclusions section is relatively brief and does not fully reflect the comprehensive findings and implications of the study. The potential long-term impact of repeated VeNT usage, whether user familiarity with the system improves over time and how participant diversity might influence short- vs. long-term usability outcomes could be considered. You may cite the studies that have tracked similar training tools over extended periods to contextualize your results and support future research directions.
Author Response
Please refer to the attached file.
For research article
|
Response to Reviewer 1 Comments
|
||
|
1. Summary |
|
|
|
Thank you very much for taking the time to review this manuscript. Please find the detailed responses below and the corresponding revisions/corrections in track changes/comments in the resubmitted manuscript files. |
||
|
2. Questions for General Evaluation |
Reviewer’s Evaluation |
Response and Revisions |
|
Does the introduction provide sufficient background and include all relevant references? |
Can be improved |
|
|
Is the research design appropriate? |
Can be improved |
|
|
Are the methods adequately described? |
Can be improved |
|
|
Are the results clearly presented? |
Can be improved |
|
|
Are the conclusions supported by the results? |
Can be improved |
|
|
Are all figures and tables clear and well-presented? |
Yes |
|
|
|
|
|
|
3. Point-by-point response to Comments and Suggestions for Authors |
||
|
|
||
|
Comments 1: While the application is named Virtual Reality Naloxone Training (VeNT), the extended version submitted for publication emphasizes mixed reality. Please clearly distinguish the original VR-based work from this new MR-based version, particularly in the Introduction and Conclusions sections. This distinction is critical to avoid confusion and to highlight the added value of MR. |
||
|
|
||
|
Response 1: Thank you for this insightful comment. In response, we have revised the manuscript to remove “virtual reality” from the intervention name. For this, we had to revise the intervention name as “ReviveXR” which is more meaningful than VeNT. It also allows expansion of the intervention in the future, because “Revive” is a broader term for life saving actions, whereas XR (i.e., extended reality) is the umbrella term for virtual reality, augmented reality, and mixed reality combined. We have made this revision n all relevant sections, including the Introduction, Methods, and Conclusion. This updated name better aligns with the current mixed reality capabilities of the application and reflects our intention to move forward with a unified identity. Additionally, we have updated the Methods section to clearly describe the MR-based implementation and differentiated it from the original VR-only design, which was the MR. We are also in the process of securing copyright under the new name to ensure clarity and consistency in future dissemination. |
||
|
|
||
|
-Comments 2: Section 2.1. Study Setting and Section 2.2. Study Design sections should be improved. What is the implementation mechanism of the iterative development process? What factors contributed to the differences between the two prototype versions in the rounds? While Section 2.3. Modifications of the Protoype addresses some of these differences, a comparison table summarizing key modifications between rounds would greatly assist reader comprehension. |
||
|
|
||
|
Response 2: We agree. We have revised sections 2.1 and 2.2 to emphasize this point. Additionally, we have included a supplementary document detailing the changes made to the program based on user input from the first round. Thank you for pointing this out. Please check the “Reviewer1 Response 2” comments in the revised manuscript – page number: 4, paragraph: 3, 4, and line: 172-17, 180; page number: 5, paragraph: 5, and line: 236-237. |
||
|
|
||
|
Comments 3: A more explicit comparison between VeNT and current standard naloxone training methods regarding efficacy and user experience would help clarify the benefits of the VeNT. Again, a comparison table could help summarize these points effectively. |
||
|
|
||
|
Response 3: Thank you for this thoughtful suggestion. We agree that a more explicit comparison between VeNT (renamed as ReviveXR) and traditional naloxone training methods would strengthen the understanding of its unique contributions. However, as this study was not designed to directly compare the two approaches, and only a few participants reported prior experience with traditional training, our comparison is limited to the qualitative reflections shared by those individuals. These insights provide useful, though anecdotal, contrasts. A more rigorous comparison with a randomized controlled trial is needed to systematically assess differences in efficacy and user experience, which we are planning as the next step. We appreciate your suggestion and see it as an important direction for future research. |
||
|
|
||
|
Comments 4: The statement, "The analysis aimed to understand the usability and acceptability of VeNT from different perspectives and reveal insights that are not immediately apparent when using quantitative analyses. " suggests that usability and acceptability are the core constructs in the study. Are there any specific, validated quantitative metrics to assess these dimensions? |
||
|
|
||
|
Response 4: Thank you for this valuable observation. The term "quantitative" was used in error; the analysis was based on qualitative methods aimed at exploring usability and acceptability from participants’ perspectives. We have corrected the wording in the revised manuscript to reflect this appropriately. We appreciate your careful review and helpful feedback. Please check the “Reviewer 1 Response 4” comments in the revised manuscript – page number: 7, paragraph: 5, and line: 297. |
||
|
|
||
|
Comments 5: For sample size & generalizability, is there any way to find an approximate sample size using an statistical analysis technique such as G-power analysis? Because the number of participants seem a bit low for the questions that must be answered in Likert scale and binary questions. Considering the diversity of participants (e.g., individuals who use drugs, bystanders, first responders, and technology specialists), a more thorough discussion of sample adequacy is warranted. Including such an analysis would help justify the generalizability of your findings. |
||
|
|
||
|
Response 5: Thank you for this important comment. We agree that a discussion of sample adequacy is essential given the diversity of participant groups. The sample size was determined a priori using G*Power, based on an alpha level of 0.05, power of 80%, and an anticipated medium effect size (d = 0.6), resulting in a minimum required sample of 19 participants. Given the relatively small overall sample size, we did not conduct subgroup analyses and instead analyzed the data as a single group. We acknowledge this as a limitation and have added clarification to the manuscript accordingly. Please check the “Reviewer 1 Response 5” comments in the revised manuscript – page number: 17, paragraph: 4, and line: 542. |
||
|
|
||
|
Comments 6: The Conclusions section is relatively brief and does not fully reflect the comprehensive findings and implications of the study. The potential long-term impact of repeated VeNT usage, whether user familiarity with the system improves over time and how participant diversity might influence short- vs. long-term usability outcomes could be considered. You may cite the studies that have tracked similar training tools over extended periods to contextualize your results and support future research directions. |
||
|
|
||
|
Response 6: We agree. We have, accordingly, revised the conclusion section to emphasize this point. Please check the “Reviewer 1 Response 6” comments in the revised manuscript – page number: 18, paragraph: 1, and line: 550-573. |
||
|
|
||
|
4. Response to Comments on the Quality of English Language |
||
|
Point 1: The English is fine and does not require any improvement. |
||
|
Response 1: No changes. |
||
|
|
||
|
5. Additional clarifications |
||
|
|
||

Reviewer 2 Report
Comments and Suggestions for Authors
The article investigates the usability of a mixed reality (MR) training tool called Virtual Reality Naloxone Training (VeNT) for educating individuals on opioid overdose response. Conducted in two phases, the study utilized the Apple Vision Pro headset and Unity platform to create immersive training scenarios. Participants (N=25) were drawn from diverse backgrounds, including drug users and first responders. Results indicated significant improvements in knowledge, confidence, and intent to intervene in overdose situations post-training. Qualitative feedback highlighted VeNT's engaging nature and its capacity to enhance learning compared to traditional training methods. The study concludes that VeNT represents a scalable, effective alternative for naloxone education, stressing the importance of practical training to bridge knowledge and real-world application gaps.
Minor Comments:
1. The abstract could benefit from clearer delineation of the study's objectives and outcomes to enhance readability.
2. Inclusion of specific data points regarding participant demographics in the abstract would provide contextual clarity.
Major Comments:
1. The discussion section could be strengthened by integrating more comparisons with existing literature on naloxone training, particularly regarding the effectiveness of MR compared to traditional methods.
2. The conclusion would benefit from a more detailed exploration of the implications for future research and practical applications, including potential barriers to implementation in community settings.
3. The phrase "an immersive and customizable experience" could be rephrased for clarity; consider specifying "which offers an immersive and customizable experience."
4. There are instances of inconsistent tense usage, particularly in the results section; ensure that all findings are consistently presented in the past tense.
5. How do the authors justify the choice of the Apple Vision Pro headset over other available VR technologies in the context of this study?
6. What specific measures were taken to assess the long-term retention of skills acquired through the VeNT program, and how were these measured?
Author Response
Please find the attached file.
For research article
|
Response to Reviewer 2 Comments
|
||
|
1. Summary |
|
|
|
Thank you very much for taking the time to review this manuscript. Please find the detailed responses below and the corresponding revisions/corrections in track changes/comments in the resubmitted manuscript files. |
||
|
2. Questions for General Evaluation |
Reviewer’s Evaluation |
Response and Revisions |
|
Does the introduction provide sufficient background and include all relevant references? |
Can be improved |
|
|
Is the research design appropriate? |
Can be improved |
|
|
Are the methods adequately described? |
Can be improved |
|
|
Are the results clearly presented? |
Can be improved |
|
|
Are the conclusions supported by the results? |
Must be improved |
|
|
Are all figures and tables clear and well-presented? |
|
|
|
|
|
|
|
3. Point-by-point response to Comments and Suggestions for Authors |
||
|
Comments 1: Minor: The abstract could benefit from clearer delineation of the study's objectives and outcomes to enhance readability. |
||
|
|
||
|
Response 1: Thank you for pointing this out. We have revised the abstract to address this comment – page number: 1, paragraph: 1, and line 21-29. |
||
|
|
||
|
Comments 2: Minor: Inclusion of specific data points regarding participant demographics in the abstract would provide contextual clarity. |
||
|
|
||
|
Response 2: Thank you. We have, accordingly, revised the abstract to emphasize this point. Please check the “Reviewer 2 Response 2” comments in the revised manuscript – page number: 1, paragraph: 1, and line: 33, 34. |
||
|
|
||
|
Comments 3: Major: The discussion section could be strengthened by integrating more comparisons with existing literature on naloxone training, particularly regarding the effectiveness of MR compared to traditional methods. |
||
|
|
||
|
Response 3: Agreed and, accordingly, we did another literature search to address your comment. However, we could not find a prior naloxone training that uses mixed reality in literature. |
||
|
|
||
|
Comments 4: Major: The conclusion would benefit from a more detailed exploration of the implications for future research and practical applications, including potential barriers to implementation in community settings. |
||
|
|
||
|
Response 4: Agree. We have, accordingly, modified the conclusion section to emphasize this point - page number: 18; paragraph: 2, lines: 50-573. |
||
|
|
||
|
Comments 5: Major: The phrase "an immersive and customizable experience" could be rephrased for clarity; consider specifying "which offers an immersive and customizable experience." |
||
|
|
||
|
Response 5: Thank you for the suggestion. We agree that the phrase could be clarified for improved readability. While a more detailed explanation of the immersive and customizable experience is provided in the following paragraph, we have revised the first paragraph to briefly clarify this point, as recommended. Please check the “Reviewer 2 Response 5” comments in the revised manuscript – page number: 4, paragraph: 5, and line: 187-190. |
||
|
|
||
|
Comments 6: Major: There are instances of inconsistent tense usage, particularly in the results section; ensure that all findings are consistently presented in the past tense. |
||
|
|
||
|
Response 6: Agree and thank you. We have accordingly modified the Result section to emphasize this point. |
||
|
|
||
|
Comments 7: Major: How do the authors justify the choice of the Apple Vision Pro headset over other available VR technologies in the context of this study? |
||
|
|
||
|
Response 7: Agree. The Apple Vision Pro is a cutting-edge headset that does not require any handheld objects to issue commands for interaction with real-world items when we apply for the grant. Given the research planned for interactive experiences involving naloxone administration, chest compressions, and rescue breathing, the Apple Vision Pro perfectly met those requirements.
|
||
|
Comments 8: Major: What specific measures were taken to assess the long-term retention of skills acquired through the VeNT program, and how were these measured? |
||
|
|
||
|
Response 8: Agreed. We have revised the limitations section.to emphasize this point – page number: 18, paragraph: 5, line: 534-548. |
||
|
|
||
|
4. Response to Comments on the Quality of English Language |
||
|
Point 1: The English is fine and does not require any improvement. |
||
|
Response 1: No changes. |
||
|
|
||
|
5. Additional clarifications |
||
|
|
||

Round 2
Reviewer 1 Report
Comments and Suggestions for Authors
The revision requests have been addressed appropriately.
Reviewer 2 Report
Comments and Suggestions for Authors
I have no other comments.